# A transcriptomic taxonomy of *Drosophila* circadian neurons around the clock

**Dingbang Ma[1†], Dariusz Przybylski[1†], Katharine C Abruzzi[1], Matthias Schlichting[1], Qunlong Li[1], Xi Long[2], Michael Rosbash[1]\***

[1]Howard Hughes Medical Institute, Brandeis University, Waltham, United States; [2]Howard Hughes Medical Institute, Janelia Research Campus, Ashburn, United States

**Abstract** Many different functions are regulated by circadian rhythms, including those orchestrated by discrete clock neurons within animal brains. To comprehensively characterize and assign cell identity to the 75 pairs of *Drosophila* circadian neurons, we optimized a single-cell RNA sequencing method and assayed clock neuron gene expression at different times of day. The data identify at least 17 clock neuron categories with striking spatial regulation of gene expression. Transcription factor regulation is prominent and likely contributes to the robust circadian oscillation of many transcripts, including those that encode cell-surface proteins previously shown to be important for cell recognition and synapse formation during development. The many other clock-regulated genes also constitute an important resource for future mechanistic and functional studies between clock neurons and/or for temporal signaling to circuits elsewhere in the fly brain.

**\*For correspondence:**
rosbash@brandeis.edu

[†]These authors contributed equally to this work

## Introduction

Eukaryotic circadian clocks rely on a now well-defined transcription-translation feedback loop, which generates roughly ~24 hr periodicity (*Takahashi, 2017*; *Top and Young, 2018*). Importantly, these circadian transcription factors (TFs) drive the oscillating expression of many thousands of tissue-specific output genes, which regulate diverse aspects of physiology and behavior (*Meireles-Filho et al., 2014*).

Circadian clocks tick away in many different mammalian tissues including the brain. The brain clock is most apparent – has very strong core clock gene expression – in a restricted region of the anteroventral hypothalamus, the suprachiasmatic nucleus (SCN) (*Reppert and Moore, 1991*). There are approximately 10,000 neurons in each unilateral mice SCN. The comparable region in *Drosophila* contains about 75 clock neurons on each hemisphere of the fly brain. The SCN in mammals and these 75 pairs of neurons in flies orchestrate many circadian functions, including behaviors (*Allada and Chung, 2010*; *Siepka et al., 2007*).

The fly brain clock neurons have been principally classified based on anatomy. There are lateral neurons and dorsal neurons. The lateral neurons are separated into three groups: ventral lateral neurons (nine LNvs), dorsal lateral neurons (six LNds), and lateral posterior neurons (three LPN). The LNvs are further subdivided into large and small LNvs based on size and location, four l-LNvs and five s-LNvs, respectively. The dorsal neurons constitute a much larger fraction of the 75 clock neurons and are also separated into four groups based on location: 15 DN1ps, 2 DN1as, 2 DN2s, and 35–40 DN3s (*Shafer et al., 2006*; *Helfrich-Förster, 2004*). Not surprisingly perhaps, these different clock neuron groups are distinct by multiple criteria. For example, calcium and functional assays indicate that different clock neurons are active at different times during a 24-hr cycle (*Liang et al., 2016*; *Guo et al., 2017*). Although microarray and RNA sequencing studies also indicate that different clock neuron groups express different sets of genes and transcripts (*Abruzzi et al., 2017*; *Kula-*

*Eversole et al., 2010*; *Nagoshi et al., 2010*), the extent of gene expression heterogeneity is often underestimated by group sequencing.

To comprehensively investigate clock neuron gene expression heterogeneity, we leveraged single-cell RNA sequencing to systematically characterize most *Drosophila* clock neurons – under light: dark (LD) and constant darkness (DD) conditions and 'around the clock'. An unsupervised clustering method confidently identified 17 clock neuron classes. These clusters were successfully mapped based on both known and novel marker genes to cover most if not all functionally distinct clock subgroups. Cell-type-specific enriched and oscillating transcripts were identified by differential gene expression. Strikingly, they include cell-surface molecules, previously shown to be important for cell recognition and synapse formation during development. We suggest that the circadian regulation of these specificity molecules orchestrate the temporal regulation of synapse formation and/or strength within the circadian network or between this network and other locations in the adult nervous system. Our study more generally establishes a foundation for neuron-specific mechanistic and functional studies within the fly brain circadian network.

## Results

### Single-cell RNA sequencing of *Drosophila* clock neurons around the clock

To systematically study *Drosophila* clock neuron gene expression at the single-cell level, flies were entrained, either in a 12:12 LD cycle or in DD, and collected at six different times around the clock. (There were also two replicates for each condition and timepoint.) The clock neuron driver *Clk856-GAL4* was used to express EGFP in most adult brain clock neurons (*Gummadova et al., 2009*), which were indeed GFP- and TIM (TIMELESS) -positive (*Figure 1A*). The number of TIM- and GFP-positive neurons were counted based on their anatomical distribution in each brain hemisphere (*Figure 1B,C*; numbers on top of the bar graph) and compared to the known number of cells in each clock neuron group in a single brain hemisphere (noted in x-axis labels, *Figure 1C*). Notably, most if not all neurons in every group were GFP-positive; the only exception was the DN3s with 30% GFP-positive neurons. Cell-specific knock-down of critical clock components with *Clk856-GAL4* mimics the phenotypes of classic circadian mutant lines (*Schlichting et al., 2019*), consistent with the notion that the GFP-positive cells identify many functionally important clock neurons. There were also approximately seven ectopic GFP-positive cells in each hemisphere; they were TIM-negative by immunohistochemistry (*Figure 1A–C*).

Since *Clk856-GAL4* driving EGFP successfully identified most circadian neurons, we isolated them using fluorescence-activated cell sorting (FACS) and generated single-cell sequencing libraries using a method based on the CEL-Seq2 technique (*Hashimshony et al., 2016*). CEL-Seq2 is a ligation-free method, and in vitro amplification mitigates possible PCR bias. This approach detects more genes, especially weakly expressed genes, than many other methods (*Mereu et al., 2020*). We optimized this method for *Drosophila* clock neurons (*Figure 1—figure supplement 1A*) by using two rounds of in vitro transcriptions to amplify polyA RNA (*Paul et al., 2017*). Each cell was sequenced to a depth of about 0.5 million reads, and approximately 2300 cells from LD- and DD-entrained flies passed the stringent filtering criteria for a total of 4671 cells (*Figure 1—figure supplement 1B*). There were about 2600 genes detected per cell (mean) with an average of about eight UMIs (unique molecular identifiers; about 20,000 UMIs total). This indicates deep molecular characterization as only a few genes and UMIs were detected in empty wells (*Figure 1—figure supplement 1C*). We pooled the cells collected at different times of day based on the replicates and LD cycles to reconstruct pseudo-bulk samples. The Pearson correlation between the corresponding replicates demonstrates that the sequencing data are highly reproducible, and even the Pearson correlation between LD and DD is 0.97, indicating highly similar gene expression between these two conditions (*Figure 1—figure supplement 1D*). This tight correlation likely emphasizes housekeeping transcripts as well as the averaging of the time point data.

We next integrated together the cells collected in LD and DD conditions cross different times of day by unsupervised clustering with Seurat (*Butler et al., 2018*), which resulted in 39 distinct clusters (*Figure 1D*). Every cluster contains cells from all six time points and from both LD- and DD-entrained flies with only modest variability (*Figure 1—figure supplement 1E,F*). There is, however, much more

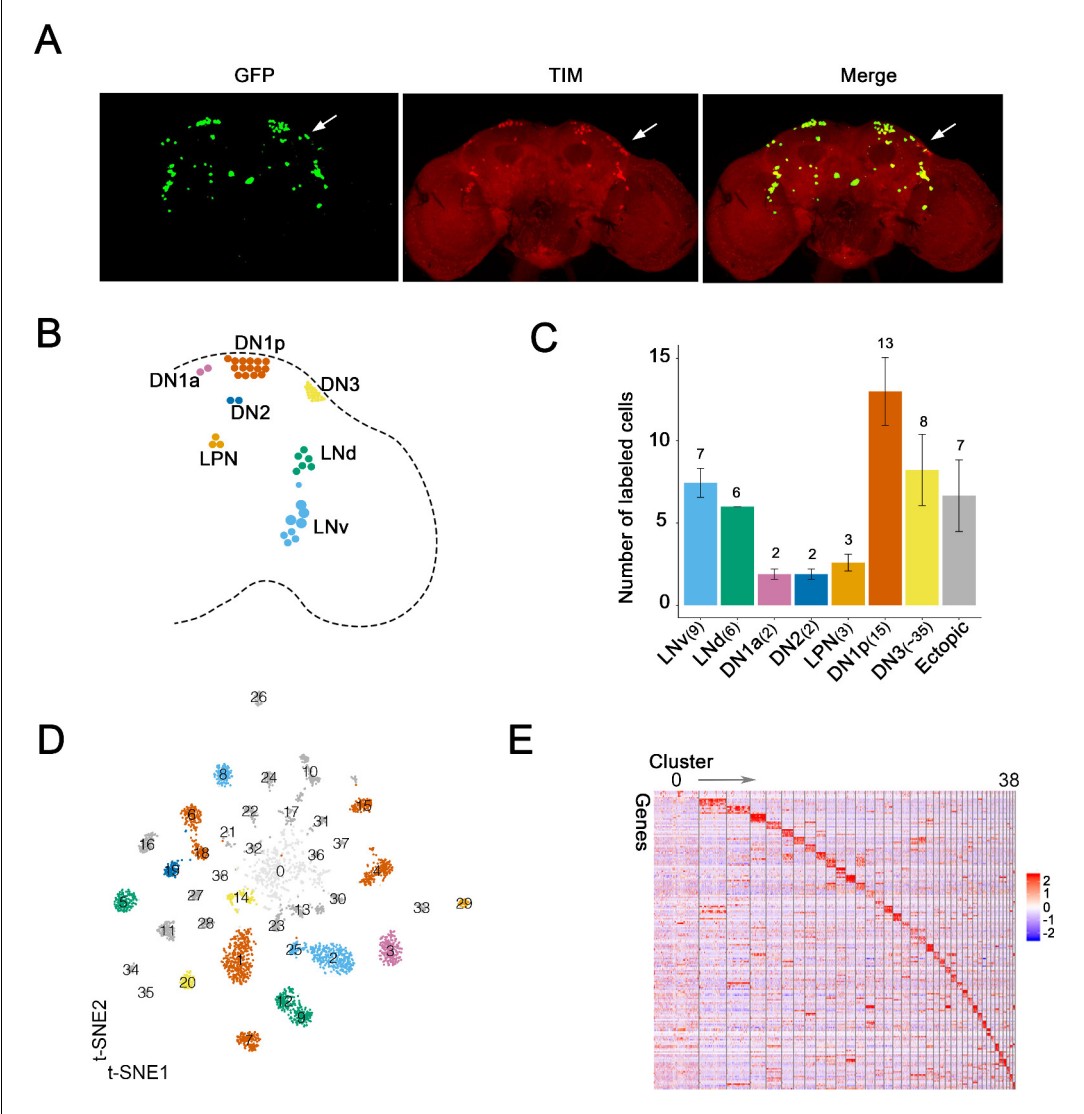

**Figure 1.** Single-cell RNA sequencing of *Drosophila* clock neurons around the clock. (**A**) Confocal stack images of immunostained brains from *Clk856-GAL4 > UAS-Stinger-GFP* flies at ZT18. Anti-GFP (left), anti-TIM (middle) and a merge of these two images (right). The arrow indicates the DN3 neurons. (**B**) Schematic depiction of clock neurons in an adult *Drosophila* brain. The core clock network consists of about 150 lateral and dorsal neurons, they are subdivided into different groups based on anatomy. (**C**) Quantification of the neurons labeled by *Clk856-GAL4* as shown in (**A**). The number on top of the bars represent the number neurons that were observed to be GFP- and TIM-positive (clock neuron subgroups) or just GFP-positive (ectopic neurons). The numbers indicated in the x-axis labels reflect the number of each type of neuron that is found in one hemisphere of an adult fly brain. Nearly, all neurons in the clock network are labeled by the *Clk856-GAL4* driver with the exception of some of the DN3s. (**D**) Visualization of *Drosophila* clock neuron clusters on the t-Distributed stochastic neighbor embedding (t-SNE) plot. Each dot represents a cell and they are color-coded by major anatomy groups. Thirty-nine different clusters were found. Clustering contains all cells: both light:dark (LD) and constant darkness (DD) conditions at all timepoints. (**E**) Each neuron cluster is defined by a set of highly differentially expressed genes. The heatmap showing the expression levels of the top differentially expressed genes (rows) in cells (columns). Clusters are ordered by size: cluster 0 on the left and cluster 38 on the right. Expression is indicated by Z score of ln (TP10K+1); red indicates high expression and blue indicates low expression.

The online version of this article includes the following figure supplement(s) for figure 1:

**Figure supplement 1.** Methods and verification of single-cell RNA sequencing of *Drosophila* clock neurons.

variability in the number of cells per cluster, and they are numbered based on the number of cells they include: 0 has the most cells, 38 the least (***Figure 1—figure supplement 1G***). Nonetheless, there are similar numbers of genes and UMIs per cell in each cluster (***Figure 1—figure supplement***

*1H*). Moreover, every cluster except cluster 0 shows a unique marker gene expression profile, supporting the validity of even the smallest clusters (*Figure 1E*).

## Identifying the key circadian neuron clusters based on core clock gene expression

To identify clusters most likely to correspond to the well-characterized clock neuron groups, we compared *tim* as well as *Clk* expression between the 39 clusters. As expected, *tim* mRNA is cycling with a gene expression peak at ZT14-ZT18 in most if not all clusters (*Figure 2A*). The data for *vri* mRNA is similar (*Figure 2—figure supplement 1*). There is, however, a marked difference in the *tim* and *vri* expression levels per cell among the different clusters. Despite a few exceptional clusters, those with more cells generally show considerably higher clock gene expression per cell (*Figure 2A*, *Figure 2—figure supplement 1*). *Clk* expression at its usual LD peak time of ZT2 is similarly higher in almost all of these same clusters (*Figure 2B*). Based on the notion that higher levels of *Clk* and CLK-CYC direct target genes may identify bona fide circadian clock neurons and to focus on a smaller number of clusters, we chose 17 for analysis in more detail (*Figure 2*; purple boxes). They retain their numbers from their position in the original 39 clusters, and many of their names carry additional labels (*Figure 3A*) to reflect the information described below.

## Identification of the lateral neuron clusters: LNvs and LNds

To identify lateral neuron clusters, we first addressed the expression of the famous circadian neuropeptide gene *Pdf*. It is expressed in just eight clock cells and two cell types, the four large and four small LNvs on each side of the brain; they are classically referred to as l-LNvs and s-LNvs, respectively (*Helfrich-Forster, 1995*). *Pdf* is expressed in only two clusters, 2 and 25 (*Figure 3B*); cluster 2 but not cluster 25 also expresses substantial levels of *sNPF* mRNA (*Figure 3C*). As sNPF is expressed in s-LNvs but not l-LNvs (*Johard et al., 2009*), these data suggest that cluster 2 identifies s-LNvs and

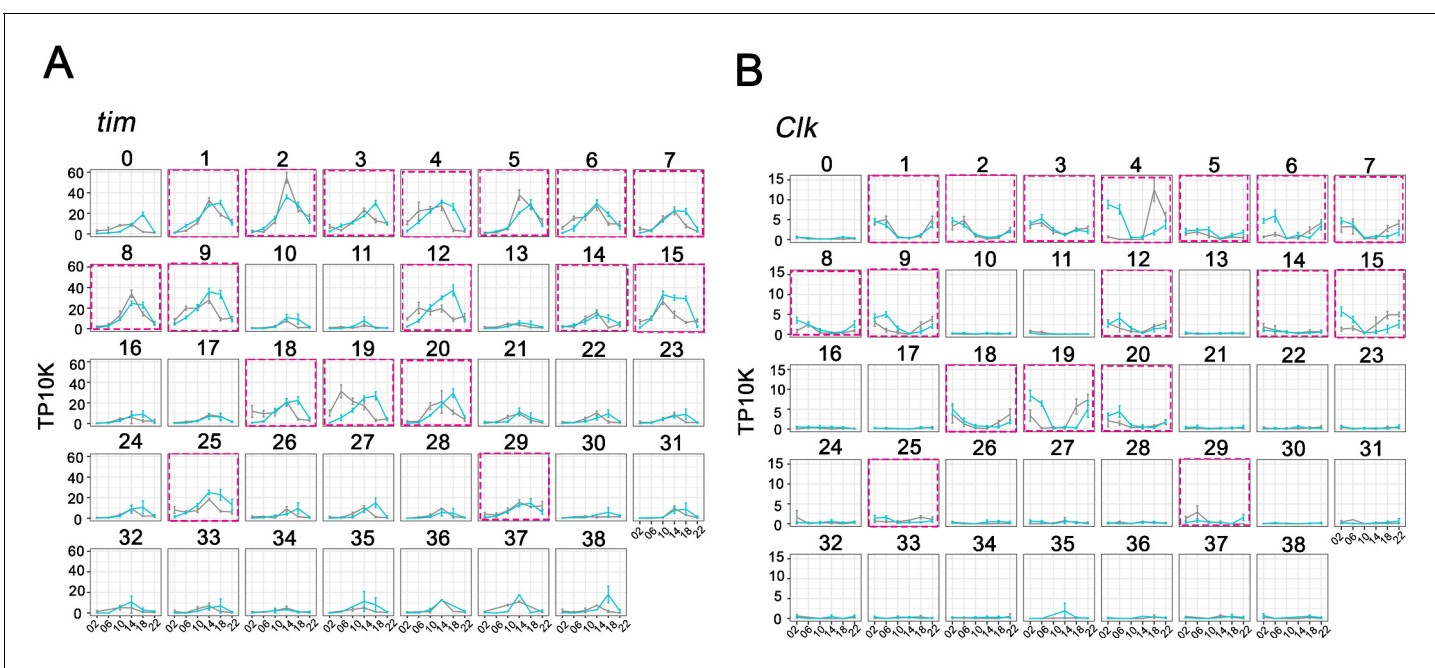

**Figure 2.** Cycling *timeless* (*tim*) expression and *Clock* (*Clk*) abundance defines key circadian neuron clusters. (**A–B**) The mean *tim* expression (**A**) and *Clk* (**B**) throughout the day in light: dark (LD) and constant darkness (DD) conditions is graphed for each cluster. Seventeen clusters were chosen as high confidence clusters based on robust core clock genes expression (purple boxes). X axis indicates the time points in LD and DD. Error bars represent mean ± SEM. Cyan and gray lines indicate the gene expression in LD and DD conditions, respectively.

The online version of this article includes the following figure supplement(s) for figure 2:

**Figure supplement 1.** Cycling *vrille* (*vri*) expression helps to define key circadian neuron clusters.

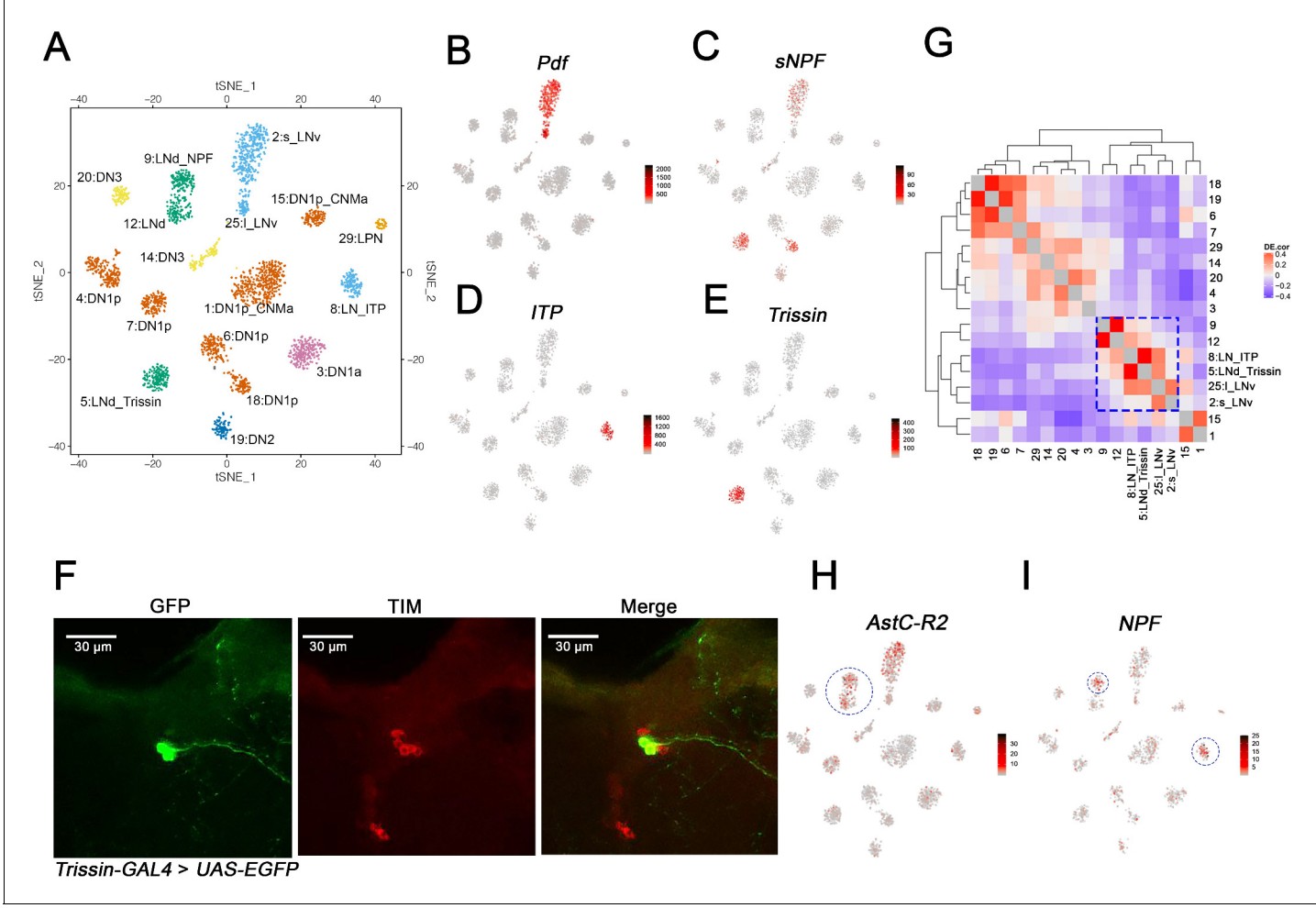

**Figure 3.** Identification of the lateral neuron clusters: LNvs and LNds. (A) Seventeen *Drosophila* clock neuron clusters. t-SNE visualization of 2615 clock neurons from both light:dark (LD) and constant darkness (DD). Each cluster retains its original identifying number (as in *Figure 1D*), and the anatomical cluster to which it was assigned is added to its name. The clusters are colored by their major anatomy groups. (B–E) t-SNE plots showing key lateral neuron marker genes expression: *Pdf* (B), *sNPF* (C), *ITP* (D) and *Trissin* (E). Red indicates higher expression (color bar, TP10K), LD and DD data are plotted together. (F) Confocal stack images from *Trissin-GAL4 > UAS-EGFP* adult fly brains stained with GFP and TIM antibodies. GFP (left), TIM (middle), and a merge of these two images (right). Only two neurons in the clock network were identified using the *Trissin-GAL4* knock-in line. (G) Gene expression correlation of the identified clock neuron clusters. (H–I) t-SNE plots showing key lateral neuron marker genes expression: *AstC-R2* (H) and *NPF* (I). Red indicates higher expression (color bar, TP10K). *AstC-R2* expression is from LD and DD together, *NPF* expression is from LD condition only. The online version of this article includes the following figure supplement(s) for figure 3:

**Figure supplement 1.** Identification of two Trissin-expressiong LNd neurons.

cluster 25 identifies l-LNvs. The s-LNv cluster 2 has many more cells than the l-LNv cluster 25 despite the fact that each fly brain has the same number of s-LNvs and l-LNvs (see Discussion).

To address the remaining seven lateral neurons, the six LNds and the (PDF-negative) fifth s-LNv, we focused on neuropeptides known to be expressed in these neurons. One LNd and the fifth s-LNv express ion transport peptide (ITP) (*Johard et al., 2009*; *Schubert et al., 2018*). As *ITP* mRNA identifies only a single cluster, #8 (*Figure 3D*), we named it LN_ITP and suggest that it contains these two cells, the fifth s-LNv and this single LNd. A single cluster implies that the transcript composition of these two neurons is very similar, consistent with their indistinguishable functional and connectivity properties (*Yao and Shafer, 2014*).

The neuropeptide sNPF is expressed in two LNd cells in addition to the s-LNvs (*Johard et al., 2009*), and this transcript is well-expressed in clusters 5 and 18 as well as in the s-LNv cluster 2 as described above (*Figure 3C*). Cluster 18 is identified as a dorsal neuron cluster (see below). Cluster

5 has prominent expression of another neuropeptide, *Trissin* (*Figure 3E*). A chemoconnectome *Trissin* endogenous *GAL4* knock-in line (*Deng et al., 2019*) resulted in expression in two LNds (*Figure 3F*), and an intersectional assay combining *Trissin-LexA* with *Clk856-GAL4* also expressed only in these two LNds (*Figure 3—figure supplement 1*). In both these experiments, the TIM immunostaining signal co-localized with GFP and further confirmed that these neurons are LNds (*Figure 3F*, *Figure 3—figure supplement 1*). Based on these data, we named this two-cell cluster 5: LNd_Trissin.

To identify the remaining three LNds, we examined gene expression similarity to find clusters most similar to the two previously defined lateral neuron clusters, LN_ITP and LNd_Trissin. These are clusters 9 and 12 (*Figure 3G*; blue box). The transcript encoding the AstC receptor, *AstC-R2* mRNA is well-expressed in these two clusters (*Figure 3H*) and in LNds based on previous RNA sequencing and physiology experiments (*Abruzzi et al., 2017*; *Díaz et al., 2019*). Because NPF is also expressed in two LNds that are *AstC-R2* positive (*Lee et al., 2006*), this explains the presence of *NPF* in cluster 9, which must contain these two cells. We noticed *NPF* is upregulated in most of the clusters in DD, for cluster classification, *NPF* expression in LD condition is shown in *Figure 3I*. By elimination, cluster 12 must contain the remaining single LNd. Therefore, we have named these two clusters 9: LNd_NPF and 12: LNd.

## Identifying the dorsal and LPN clock neurons

Dorsal clock neurons (DNs) constitute the majority of *Drosophila* clock neurons, but their organization and function—how many different cell types there are and what they do—are far from certain. There are, however, some small DN subgroups about which more is known (*Shafer et al., 2006*).

The two DN1a neurons are the only source of CCHa1 in the fly clock system (*Fujiwara et al., 2018*), and immunohistochemistry of a chemoconnectome CCHa1 line indicates that this gene is indeed expressed in only two clock neurons at the expected location for DN1as (*Figure 4—figure supplement 1A*). As only cluster 3 expresses high levels of CCHa1, we assigned the two DN1as to this cluster (*Figure 4A*). For DN2 neuron assignment, we took advantage of the fact that DN2 molecular oscillations are in anti-phase with the rest of the clock neuron population in DD (*Kaneko et al., 1997*). Remarkably, only cluster 19 shows unambiguous anti-phasic DD cycling for a number of core clock genes including *tim* (*Figure 2* and *Figure 4B*) and *vri* (*Figure 2—figure supplement 1*), justifying the assignment of the DN2s to cluster 19. The three LPNs are the only clock neuron group with both *AstC* and *AstA* expression (*Díaz et al., 2019*; *Ni et al., 2019*); only the small cluster 29 meets this criterion (*Figure 4C,D*).

A number of criteria were used to identify the 15 DN1p neurons and their surprising assignment to six different clusters. First, the *glass* (*gl*)-encoded TF is expressed in and necessary for the development of most DN1ps (*Helfrich-Förster et al., 2001*; *Shafer et al., 2006*). Five novel clusters (1, 6, 7, 15, and 18) express the *gl* transcript, suggesting that they contain DN1ps (*Figure 4E*; five of the six circles). Cluster 18 is *sNPF* enriched as shown above (*Figure 3C*). Second, two of these five clusters, clusters 1 and 15, express the transcript encoding the neuropeptide Dh31 (*Figure 4F*). Previous results indicate that *Dh31* is expressed in some DN1ps, and loss- and gain-of-function studies of *Dh31* indicate that its expression within this set of clock neurons regulates fly sleep (*Kunst et al., 2014*; *Goda et al., 2016*). Third, these same two clusters also express the transcript encoding the neuropeptide CNMa (*Figure 4G*). To verify the assignment of these two CNMa clusters, we used a chemoconnectome knock-in *CNMa-GAL4* to drive EGFP for immunohistochemistry. The only TIM-positive EGFP-positive neurons were in the position of DN1ps (*Figure 4I*). These six to eight EGFP-positive dorsal cells overlapped well with the well-known DN1p Clk4.1 driver (*Clk4.1M-LexA*), indicating that these two DN1p clusters contain about half the DN1ps (*Figure 4J*). Fourth, expression of the neuropeptide gene *AstC* mRNA identified only a single DN1p cluster, the larger *Dh31*-positive and *CNMa*-positive cluster 1 (*Figure 4D*), an assignment that is consistent with recent work indicating that *AstC* is expressed in a subset of DN1ps (*Díaz et al., 2019*).

Lastly, *Rh7* expression identified an additional 6th DN1p cluster, cluster 4 (*Figure 4E,H*). *Rh7* mRNA is present in this cluster as well as three others: the DN1as, DN2s, and the *gl*-positive DN1p cluster 7 (*Figure 4H*). This assignment of cluster 4 to the DN1ps is despite the lack of robust *gl* expression and based in part on previous characterization of a *Rh7* mimic line. It indicated that *Rh7* may be only expressed in four to five DN1ps as well as the two DN1as (*Kistenpfennig et al., 2017*).

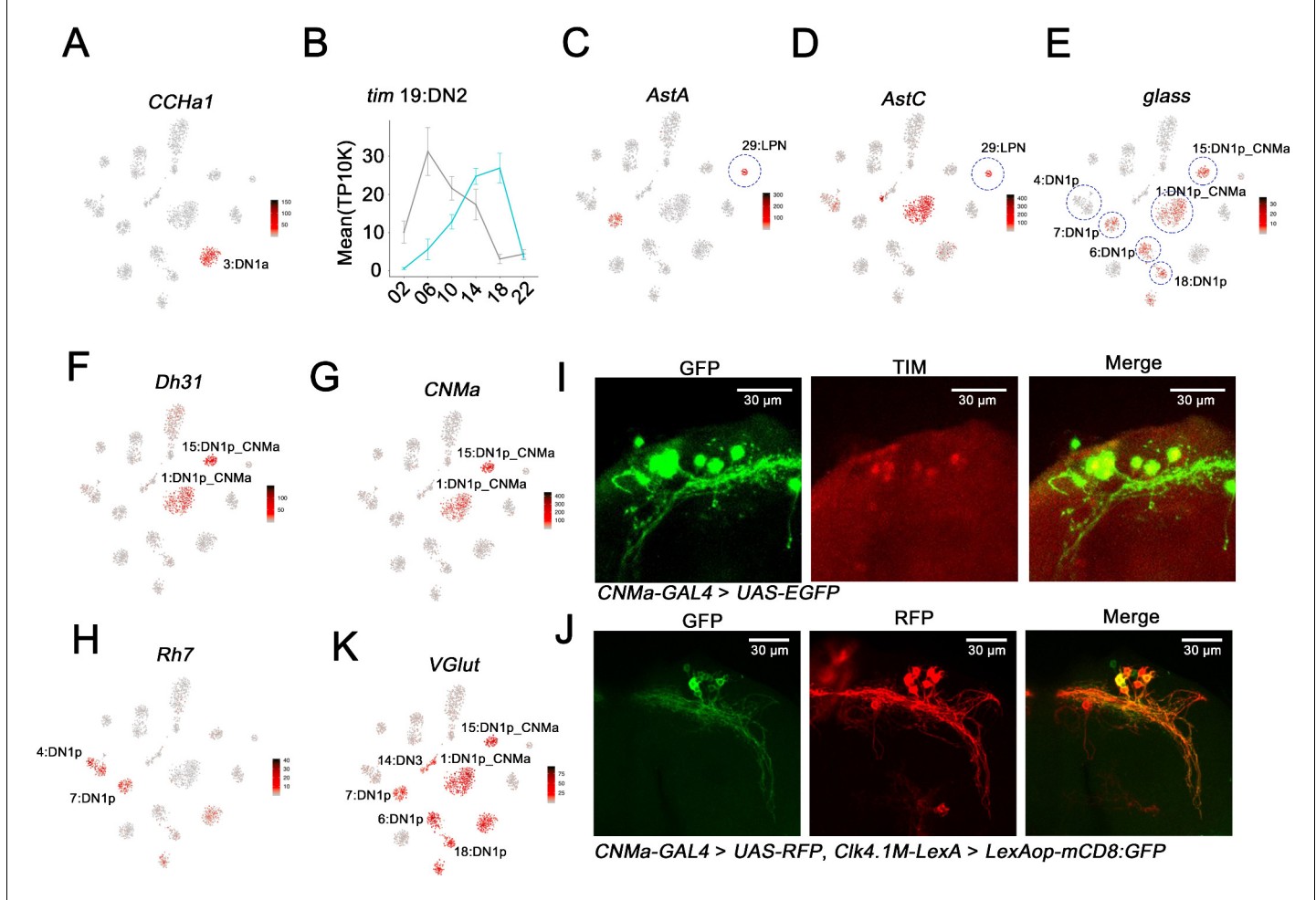

**Figure 4.** Mapping the dorsal and lateral posterior neuron (LPN) clock neurons. (**A**) t-SNE plots showing *CCHa1* expression. Red indicates higher expression (color bar, TP10K). Light:dark (LD) and constant darkness (DD) data are plotted together. *CCHa1* expression is specific to cluster 3 and thus identifies the DN1a neurons. (**B**) *tim* expression in cluster 19 in LD and DD conditions. The phase of *tim* cycling shifts in DD to be approximately antiphase to LD cycling. DN2s assigned to Cluster 19. Error bars represent mean ± SEM. Cyan and gray lines indicate the gene expression in LD and DD conditions, respectively. (**C–D**) t-SNE plot showing *AstA* (**C**) and *AstC* (**D**) expression. Red indicates higher expression (color bar, TP10K). LD and DD data are plotted together. The LPN cluster is highlighted by the blue circle. (**E–H**) t-SNE plot showing *glass* (**E**), *Dh31* (**F**), *CNMa* (**G**), and *Rh7* (**H**) expression. Red indicates higher expression (color bar, TP10K); LD and DD data are plotted together. The identified DN1p neuron clusters are highlighted by blue circles showing in (**E**). (**I**) Confocal stack of images showing antibody staining for GFP (left) and TIM (middle) and the merge (right) in the dorsal brain of *CNMa-GAL4 > UAS-EGFP* flies. (**J**) Confocal stack of images showing antibody staining for GFP (left) and RFP (middle) and the merge (right) in the dorsal brain of *Clk4.1M-LexA > LexAop-mCD8: GFP; CNMa-GAL4 > UAS-RFP* flies. (**K**) t-SNE plot showing *VGlut* expression. Red indicates higher expression (color bar, TP10K), LD and DD data are plotted together.

The online version of this article includes the following figure supplement(s) for figure 4:

**Figure supplement 1.** Mapping the dorsal clock neurons.

To further assess *Rh7* expression, we generated a split-GAL4 line using regulatory regions of *per* and *Rh7*; functional GAL4 should only be expressed in clock cells that also express *Rh7*. Consistent with the *Rh7* mRNA expression pattern in the clusters, this assay also indicated *Rh7* expression in two DN1ps and the DN2 neurons (*Figure 4—figure supplement 1B*).

The glutamate vesicular transporter mRNA *VGlut* is enriched in most dorsal neuron clusters (*Figure 4K*). Consistent with this conclusion is a split-GAL4 assay with *Pdfr* and *VGlut* regulatory regions: it indicates that most DN1ps express *VGlut* (*Figure 4—figure supplement 1C*); the RNA profiling indicates that only a single DN1p cluster—the *gl*-negative *Rh7*-positive cluster 4—is an exception. These data are also consistent with prior studies indicating that some dorsal neurons are

VGlut-positive, and glutamate expression is relevant to sleep-wake behavior (*Guo et al., 2016*; *Guo et al., 2017*; *Hamasaka et al., 2007*). In summary, six DN1p clusters were identified by the known and newly identified markers (*Figure 4—figure supplement 1D*).

The two final clusters, 14 and 20, are missing almost all marker genes identified in the better characterized clock neurons and were therefore assigned to the enigmatic DN3 population. The one exception is *VGlut*. It is expressed in Cluster 14, and previous studies have shown that some DN3s are glutamatergic (*Hamasaka et al., 2007*). Gene expression correlation analysis indicates that cluster 20 is similar to cluster 14 (*Figure 3G*), so we also assigned this last cluster to the DN3s.

## Enriched and cycling transcripts in the clock neuron clusters

To further characterize the gene expression profiles of these clock neuron clusters, we first addressed differential (enriched) gene expression. Each cluster shows a unique combination of marker gene expression (*Figure 5—figure supplement 1A,B*, *Source data 1*). Notably, TFs are prominently expressed in the clock clusters and extend well beyond the known clock gene TFs, for example *Figure 2*. Interestingly, some TF mRNAs are enriched in all clusters from a particular anatomical neuron group (*Figure 5A*). For example, *ham*, *D* as well as *gl* transcripts are enriched in most DN1p clusters (e.g. *ham Figure 5B*). Most striking is the cluster-specific TF expression in the DNs and LPNs (labeled in yellow). Each cluster has at least one TF mRNA enriched relative to the rest of the circadian network (*Figure 5A*, *Figure 5—figure supplement 1C*). For example, cluster 4 DN1s show strong expression of transcripts encoding *slou* and *so* and the TF mRNAs *hth*, *bi*, *toy*, and *Hr51* are enriched in all lateral neurons (e.g. *toy* and *Hr51*; *Figure 5C,D*). *opa* mRNA is specifically enriched in the LNvs (*Figure 5E*).

Interestingly, *CrebA* mRNA is highly expressed in only some s-LNv cluster cells (*Figure 5F*; circle), due to robust oscillations under both LD and DD conditions (*Figure 5G*). For unknown reasons, the amplitude of *CrebA* expression is more robust in DD than in LD. *pdm3* mRNA oscillates in the DN1p_CNMa cluster 1 with peak expression at midday (*Figure 5H,I*).

To address cycling gene expression more generally, we examined the entire single-cell RNA sequencing dataset for oscillating transcripts (cycling RNAs or cyclers) under both LD and DD conditions. The following cycling cutoffs were used: a JTK cycle Benjamini-Hochberg corrected q-value of less than 0.05, a F24 score of greater than 0.5, and a cycling amplitude (maximum expression divided by minimum expression) of at least 1.5-fold, and a maximal expression of at least 0.8 TP10K. Although these conservative criteria were used to mitigate against false positive (see Materials and methods), ~24% of all transcripts identified in the 17 clock neuron clusters (1320/5517) undergo time-of-day oscillations in at least one cluster (*Figure 6—figure supplement 1A*). Notably, even under the more cycling-permissive LD conditions, 74% of the LD cycling transcripts are found in only one or two clusters. This specificity may be due in part to the difficulty in identifying cyclers from the smaller clusters (see Discussion). However, even the two largest clusters (Clusters 1:DN1p_CNMa and 2_sLNv) have only ~30% overlap. In contrast, the core clock gene transcripts are identified as cyclers in most of the 17 clusters in LD (*Clk* (11), *tim* (17), *vri* (12), and *Pdp1* (13)). Several other transcripts also cycle in multiple clusters including the mRNA-binding protein *fne*, the arginine kinase *ArgK*, and the unknown transcript *CG15628* (*Figure 6—figure supplement 1B* and *Supplementary file 1*).

To address possible functions of the clock neurons cycling program, we first performed gene ontology (GO) term analysis on LD cyclers from the individual clusters (*Supplementary file 2*). This analysis was more informative for the larger clusters, as they had more cyclers than the smaller clusters. Although there are also significant GO terms shared between cyclers from different clusters such as behavior, neuron projection morphogenesis and G-protein-coupled signaling pathways, many of the GO terms indicate that cycling transcripts contribute to different processes in the different clusters. For example, the DN1p_CNMa cyclers (cluster 1) are enriched for chemical synaptic transmission (0.008), whereas the s-LNv cyclers (cluster 2) are enriched for cation membrane transport (2.1E-7).

We also performed GO term analysis on all cycling transcripts identified in any of the 17 clusters (*Supplementary file 3*). In addition to the GO terms identified in multiple individual clusters (*Figure 6A*, Asterix), several new GO terms emerged from the combined cyclers; they included second messenger-mediated signaling, cell-cell adhesion via plasma membrane adhesion molecules and axonogenesis (*Figure 6A*). These new functions required the pooling of cycling transcripts

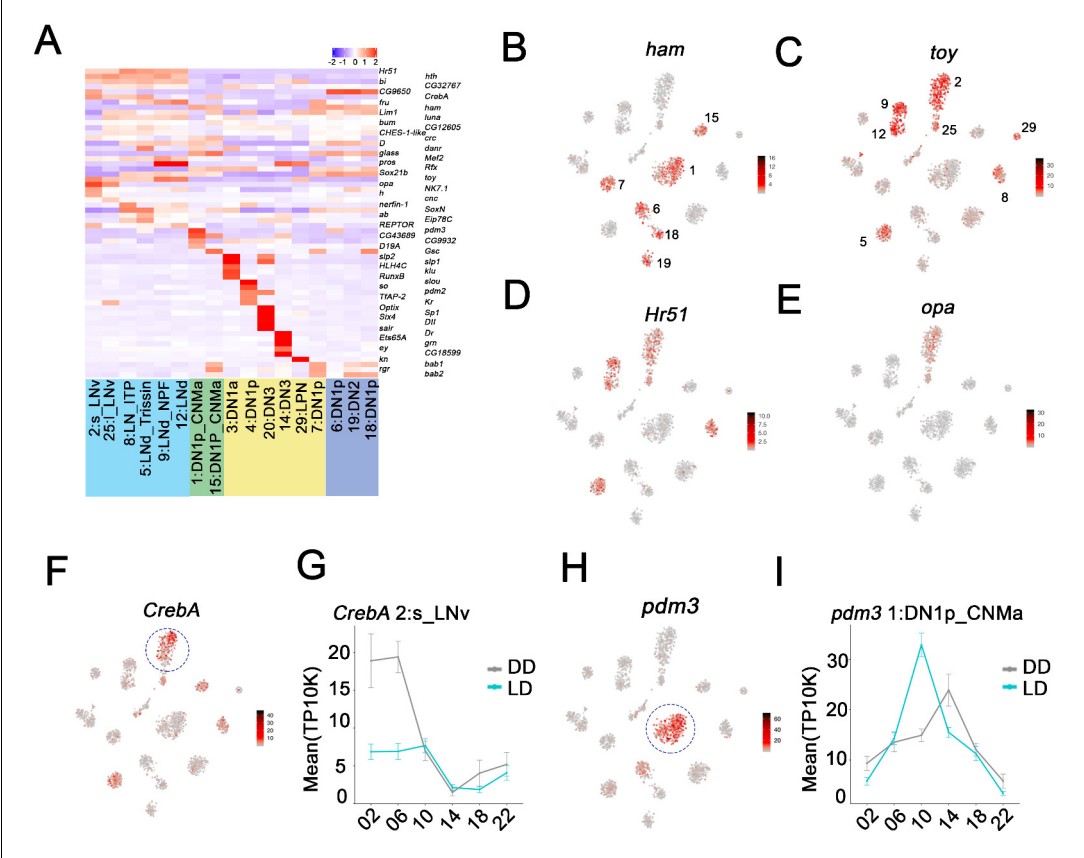

**Figure 5.** Enriched transcripts in identified clock neuron clusters. (A) Heatmap showing the enriched transcription factors expression in each cluster. (B–E) t-SNE plot showing *ham* (B), *toy* (C), *Hr51* (D), and *opa* (E) expression in clock neurons. Red indicates higher expression (color bar, TP10K). (F) t-SNE plot showing *CrebA* expression in clock neurons. Red indicates higher expression (color bar, TP10K). (G) Mean *CrebA* expression in 2:s_LNv neurons throughout the day in light:dark (LD) and constant darkness (DD) conditions. Error bars represent mean ± SEM. Cyan and gray lines indicate the gene expression in LD and DD conditions, respectively. (H) t-SNE plot showing *pdm3* expression in clock neurons. Red indicates higher expression (color bar, TP10K). (I) Mean *pdm3* expression in 1:DN1p_CNMa neurons at different times in LD and DD conditions. Error bars represent mean ± SEM. Cyan and gray lines indicate the gene expression in LD and DD conditions, respectively.

The online version of this article includes the following figure supplement(s) for figure 5:

**Figure supplement 1.** Enriched transcripts in identified clock neuron clusters.

because each clock neuron cluster only contains one or two cycling transcripts from these large gene families. This small number is insufficient to identify the function from individual clusters. To try to learn a bit more about the importance of these GO terms relative to time of day, we examined whether these GO terms were specific to cyclers peaking in the day or night (*Supplementary file 3*). Interestingly, in LD, G-protein-coupled receptor signaling was identified as a GO term only for cycling transcripts that peak at night. In contrast neuron-projection morphogenesis was identified as a GO term only for cycling transcripts peaking during the day.

Two prominent examples of the axonogenesis category are the DIP (Dpr interacting proteins; 11 genes) and Dpr (Defective proboscis extension response; 21 genes) families. A DIP protein on the surface of a neuron has affinity for partner Dpr proteins, which together help drive synapse specificity during development (*Cosmanescu et al., 2018*). Four of 11 DIP transcripts are specifically enriched and undergo LD cycling in specific clock neuron clusters (*Figure 6B*). Similarly, 10 of 21 *Dpr* transcripts are enriched and undergo LD cycling in specific clusters (*Figure 6—figure supplement 1C*). Intriguingly, *DIP-beta* mRNA is enriched and cycles strongly in s-LNvs (*Figure 6C,D*), whereas a *Dpr8* transcript encodes one of its partner proteins and cycles with a similar phase in DN1ps (*Figure 6E*). Perhaps, cycling gene expression regulation of *DIP-beta* and *Dpr8* contribute to the circadian regulation of the known interaction between s-LNvs and DN1ps. Circadian regulation of an

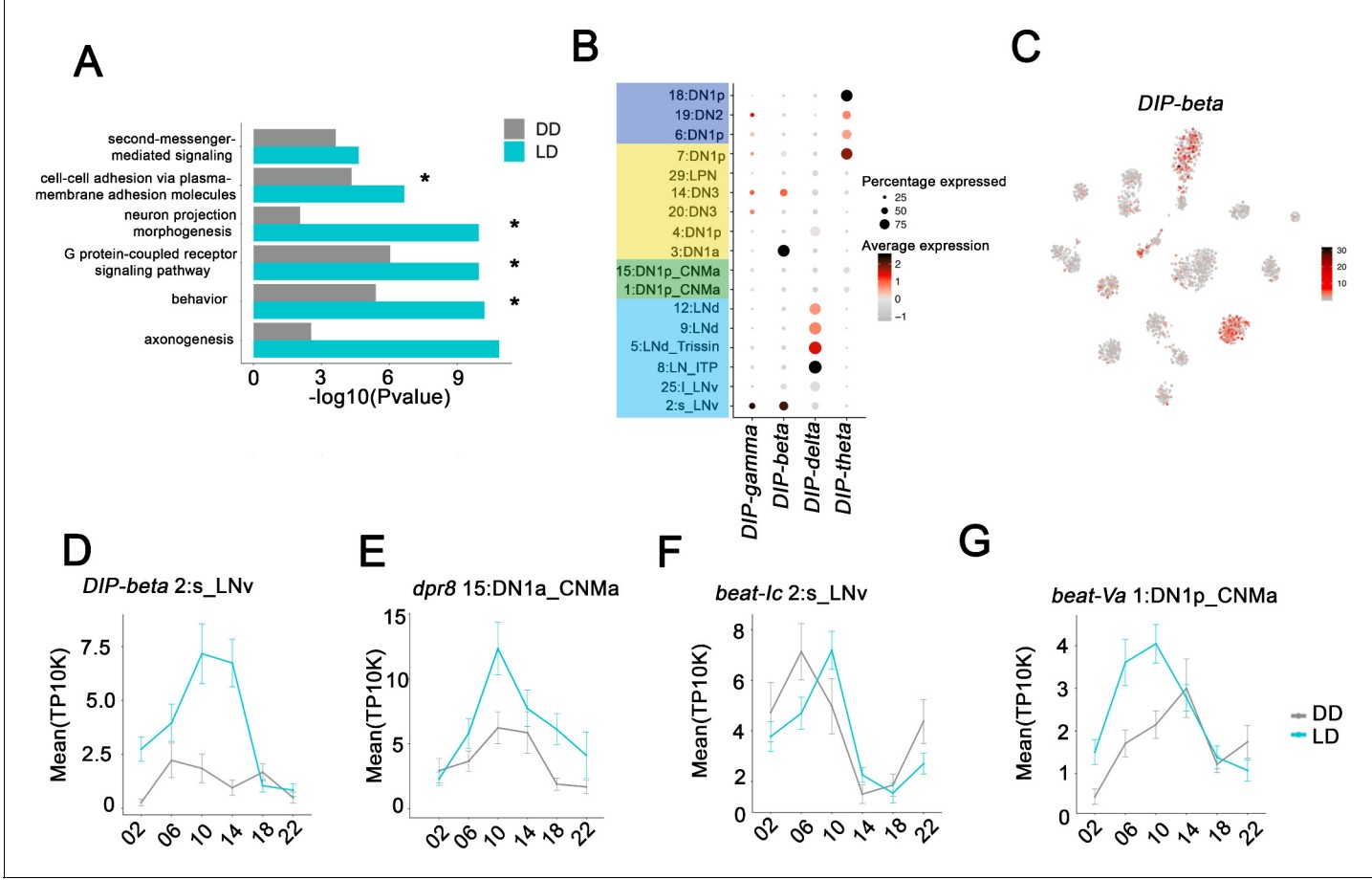

**Figure 6.** Neuron-specific cycling gene expression. (A) Gene ontology (GO) analysis of the all the cycling genes found in all the clock neuron clusters. GO terms that were also identified in cluster-specific GO term analysis are indicated by asterisks. (B) Dot plot showing the Dpr interacting protein (DIP) family gene expression in clock neuron clusters. Size of dot indicates what percentage of cells in a particular cluster that express the indicated DIP-family member. Color indicates the mean expression within that cluster. (C) t-SNE plot showing *DIP-beta* expression in clock neurons. Red indicates higher expression (color bar, TP10K). (D–G) Mean cell-cell adhesion gene expression throughout the day in different neuron clusters. *DIP-beta* expression in 2:s_LNv (D), *dpr8* expression in 15:DN1p_CNMa (E), *beat-Ic* expression in 2:s_LNv (F) and *beat-Va* expression in 1:DN1p_CNMa (G). Error bars represent mean ± SEM. Average gene expression shown in constant darkness (DD) (gray) and light:dark (LD; cyan).

The online version of this article includes the following figure supplement(s) for figure 6:

**Figure supplement 1.** Oscillating transcripts in *Drosophila* clock neurons.

interaction between s-LNv termini and a target like DN1ps was the original suggestion for the function of s-LNv morphological cycling (*Fernández et al., 2008*; *Petsakou et al., 2015*).

*Beat* transcripts encode another Ig superfamily of molecules, which are involved in axonogenesis (*Pipes et al., 2001*) and show impressive cluster-specific enrichment and cycling (*Figure 6F,G*). We speculate that all three classes of cell surface molecules help temporally coordinate synaptic contacts, principles that function broadly within the circadian circuit and even elsewhere within the brain to superimpose time-of-day control on synaptic strength if not choice (*Guo et al., 2018*).

## Discussion

*Drosophila* adult brain clock neurons are known to be heterogeneous, based principally on their anatomy, immunohistochemistry, and functional characterization (*Top and Young, 2018*; *Tataroglu and Emery, 2015*; *Peschel and Helfrich-Förster, 2011*). To extend this characterization, we optimized a single-cell RNA sequencing protocol and applied it to most of the clock network,

about 45 of the 75 neurons on each side of the brain. Only the numerous and poorly characterized DN3 clock neurons were mostly missing from the 45 assayed neurons.

We suspect that the identification of the 17 clock clusters reflects gene expression heterogeneity revealed by our modified CEL-Seq2 method. A large number of genes and UMIs resulted, which probably allowed for a more thorough definition of clock neuron cell type. The only fly brain neuron studies with similar sequencing depth per cell also reported comparable molecular heterogeneity (*Li et al., 2020*; *Li et al., 2017*).

The initial unsupervised clustering results were surprising. Assuming the immunostaining revealed all GFP-positive neurons (*Figure 1A*), the 39 clusters indicates about 1–2 cells/cluster from each side of the brain at about 8X coverage for each cell at each time point (*Figure 1D*). One possibility is that many of the small clusters are DN3 neurons that were inefficiently recovered during FACS sorting, due perhaps to lower *Clk* expression (*Figure 2B*).

The 17 final clock neuron groups contain a somewhat larger number of cells per cluster, about two to three, many of which are well-defined by immunohistochemistry and contain the predicted number of cells. They include for example the two DN1as, two LNd_Trissin cells, two DN2s, and the four s-LNvs. There are, however, clusters that contain too few cells, most notably the l-LNvs and the LPNs; they should contain four and three cells, respectively. As the correct cell numbers of l-LNvs and LPNs are easily visible by GFP-staining *of Clk856-GAL4 > UAS-Stinger-GFP* brains, many of these neurons are probably poorly recovered or perhaps incorrectly assigned.

The clustering and analysis clarified several clock neuron identity issues for the LNds and the DN1ps. There is only one *ITP*-expressing cluster, which must contain the ITP-positive LNd as well as the fifth s-LNv. Their indistinguishable gene expression profiles fit nicely with recent neuroanatomical studies suggesting that they have nearly identical properties; they innervate the same brain areas with similar pre- and postsynaptic sites, and they both drive evening activity (*Schubert et al., 2018*; *Hermann-Luibl et al., 2014*). Most surprising perhaps was the division of the DN1ps into six clusters. Two of these express the recently characterized rhodopsin gene *Rh7*, and two others the neuropeptides CNMa and Dh31. All the CNMa cells reside close together in the dorsal brain and are spatially distinct from the rest of the DN1p population, yet they are still divided into two clusters based on this molecular profiling. Two recently identified sleep-promoting DN1p groups show novel projection patterns and should be relevant (*Guo et al., 2018*; *Guo et al., 2016*), but we do not know which DN1p clusters they define.

The clustering results reveal remarkable heterogeneity of clock neuron gene expression. Nonetheless, additional post-transcriptional regulation might also be important for neuronal function. There are few examples where the single-cell sequencing results are inconsistent with functional results. For instance, AstC-R2 is reported to be expressed in only one LNd (*Díaz et al., 2019*), yet its mRNA is expressed in both clusters 9 and 12. We speculate that this transcript (as well as others) may undergo post-transcriptional regulation that further refines protein expression within the clock neuron network.

There is abundant circadian regulation of mRNA cycling in the 17 clock neuron clusters. It is therefore surprising that the canonical property of clock gene cycling is not present in all of them. However, *vri* and *tim* mRNA cycle in all 17 clusters by eye (also see *Figure 2*), suggesting that the lack of universal clock gene cycling reflects some data variability combined with the conservative computational criteria used to define cycling. In contrast and with only a few exceptions, most non-clock gene mRNA cycling remains highly neuron-specific by eye as well as by computation. Moreover, 24% of detected transcripts displayed rhythmic expression in only one or a small number of clusters. Although this could be influenced by the difficulty in identifying cyclers within the smaller clusters, the overlap of cycling transcripts is limited even between the larger clusters.

Although so much neuron-specific cycling in these 17 clusters of clock neurons might seem surprising, our previous bulk sequencing study of three categories of clock neurons (PDF cells, LNds, and DN1s) came to a similar conclusion (*Abruzzi et al., 2017*). The large percentage probably reflects some specific temporal as well as spatial gene expression regulation occurring within clock neurons. However, this neuron-specific control recapitulates lessons learned from mammals as well as from flies about the circadian regulation of gene expression: this regulation is highly cell- and tissue-specific with only the exception of core clock genes and a handful of others. In this context of cell type and tissue, it is interesting that the mammalian brain has the lowest number of tissue-specific cycling transcripts (*Zhang et al., 2014*). Perhaps, this is because the brain has the greatest

number of constituent cell types. Without assaying separately these cells, cell-specific cycling is invisible when assaying the complete tissue/organ. In this context, most of the cycling mRNAs identified in this study were not detected as cycling in fly head or brain mRNA (*Hughes et al., 2012*).

While this work was in progress, we became aware of a single-cell RNA sequencing study of the mouse SCN, the mammalian equivalent of the 150 fly clock neurons (*Wen et al., 2020*). Although only five neuronal subtypes were identified, this number is consistent with previous characterizations of SCN neuronal heterogeneity (*Welsh et al., 2010*). It is also possible that this heterogeneity is underestimated due to the lower sequencing depth afforded by droplet-based methods (*Wen et al., 2020*). We suspect that high-throughput commercial sequencing methods similarly underestimate neuronal heterogeneity because of limited sequencing depth. For example, they only identified 29 and 87 initial clusters in larval and adult *Drosophila* brains (*Davie et al., 2018*; *Brunet Avalos et al., 2019*).

In summary, this work reveals a surprising extent of fly brain clock neuron heterogeneity and provides a resource for future functional and mechanistic circadian studies. Although some of these features may be limited to the clock system, we suspect that neuronal heterogeneity will be more generally characteristic of fly brain circuits. This heterogeneity also helps explain why *Drosophila* has such a sophisticated behavioral repertoire despite a brain of only 100,000 neurons and suggests that many drivers that express in one or only a few neurons are likely to have substantial behavioral impact.

# Materials and methods

**Key resources table**

| Reagent type (species) or resource | Designation | Source or reference | Identifiers | Additional information |
|---|---|---|---|---|
| Genetic reagent (*D. melanogaster*) | UAS-Stinger | BDSC | RRID:BDSC_84277 | |
| Genetic reagent (*D. melanogaster*) | R14F03-p65.AD | BDSC | RRID:BDSC_70551 | |
| Genetic reagent (*D. melanogaster*) | VT029514-GAL4.DBD | BDSC | RRID:BDSC_75062 | |
| Genetic reagent (*D. melanogaster*) | R18H11-p65.AD | BDSC | RRID:BDSC_68852 | |
| Genetic reagent (*D. melanogaster*) | R51H05-GAL4.DBD | BDSC | RRID:BDSC_69036 | |
| Genetic reagent (*D. melanogaster*) | P{10XUAS-IVS-myr::GFP}attP2 | BDSC | RRID:BDSC_32197 | |
| Genetic reagent (*D. melanogaster*) | Clk856-GAL4 | *Gummadova et al., 2009* | Flybase: FBtp0069616 | |
| Genetic reagent (*D. melanogaster*) | Clk4.1M-LexA | *Guo et al., 2016* | Flybase: FBtp0093698 | |
| Genetic reagent (*D. melanogaster*) | UAS-EGFP | BDSC | RRID:BDSC_5428 | |
| Genetic reagent (*D. melanogaster*) | UAS-FRT-STOP-FRT-CsChrimson.mVenus and LexAop-FLP | *Guo et al., 2018* | | |

*Continued on next page*

*Continued*

| Reagent type (species) or resource | Designation | Source or reference | Identifiers | Additional information |
|---|---|---|---|---|
| Genetic reagent (*D. melanogaster*) | Trissin-LexA | *Deng et al., 2019* | | |
| Genetic reagent (*D. melanogaster*) | Trissin-GAL4 | *Deng et al., 2019* | | |
| Genetic reagent (*D. melanogaster*) | CCHa1-GAL4 | *Deng et al., 2019* | | |
| Genetic reagent (*D. melanogaster*) | CNMa-GAL4 | *Deng et al., 2019* | | |
| Antibody | Anti-PER Rabbit polyclonal | Laboratory of Michael Rosbash | | 1:1000 |
| Antibody | Anti-TIM Rat monoclonal | Laboratory of Michael Rosbash | RRID: AB_2753140 | 1:200 |
| Antibody | Anti-PDF Mouse monoclonal | DSHB | RRID: AB_760350 | 1:500 |
| Antibody | Anti-GFP Mouse monoclonal | Sigma-Aldrich | RRID: AB_390913 | 1:1000 |
| Antibody | Chicken anti-GFP | Abcam | RRID: AB_300798 | 1:1000 |
| Antibody | Rabbit anti-DsRed | Takara Bio USA | RRID: AB_10013483 | 1:200 |
| Antibody | Goat anti-mouse polyclonal | ThermoFisher | RRID: AB_2536185 | 1:200 |
| Antibody | Goat anti-rabbit polyclonal | ThermoFisher | RRID: AB_2576217 | 1:200 |
| Antibody | Goat anti-rabbit polyclonal | ThermoFisher | RRID: AB_2633281 | 1:200 |
| Antibody | Anti-PDF Mouse monoclonal | DSHB | RRID: AB_760350 | 1:500 |
| Software, algorithm | FIJI | https://fiji.sc/ | RRID: SCR_002285 | |
| Software, algorithm | Microsoft Office Excel | | | |
| Software, algorithm | RStudio | https://rstudio.com | RRID: SCR_000432 | Version 1.2.5033 |
| Software, algorithm | Adobe Photoshop CC | | RRID: SCR_014199 | |
| Software, algorithm | Code for clustering and rhythmic gene expression analysis | This paper | | R code *Ma and Przybylski, 2020*) |

## Fly rearing

In all the experiments, equal numbers of males and females were used. Flies were reared in standard cornmeal medium with yeast under 12:12 hr LD cycles.

## FACS sorting of circadian neurons

Flies were entrained for 3 days in 12:12 LD (for LD) and subjected to constant darkness for 3 days (for DD) prior to dissection. Time points were taken every 4 hr around the clock. Fly brains were dissected with standard protocol (*Abruzzi et al., 2017*). Briefly, 20 µM DNQX, 0.1 µM TTX, and 50 µM APV were added into the dissection saline (HEPES-KOH 9.9 mM pH7.4, NaCl 137 mM, KCl 5.4 mM, $NaH_2PO_4$0.17 mM, $KH_2PO_4$0.22 mM, glucose 3.3 mM and sucrose 43.8 mM). The brains were digested with papain (50 unit/ml, ~2 µl per brain) at room temperature for 30 min. Digestion was quenched with a fivefold volume of the SM medium (active Schneider's medium), and brains were washed twice with the ice-cold SM medium. Brains were triturated with flame-rounded 1000 µl pipette tips until most of the tissues were dissociated to single cells. The resulting cell suspension was filtered by 100 µm sieve prior to FACS sorting. The single-cell sorting was carried out with BD Melody. For single-cell sorting, single-cell sorting mode was used, cells were collected by 96-well plates or 384-well plates. Plates with sorted cells were centrifuged at 3000 *g* for 1 min at 4°C and then stored in −80°C until further processing.

## Single-cell RNA library preparation

The single-cell library prep was based on CEL-seq2 with some modifications (*Paul et al., 2017*; *Hashimshony et al., 2016*). *Drosophila* neurons were sorted into 96-well or 384-well plates prefilled with 0.6 µl primer mix; two empty wells on each plate were used as negative controls. We made four libraries from a 384-well plate. Sorted plates were centrifuged at 3000 *g* for 1 min at 4°C, and then stored in −80°C. Eppendorf EPmotion liquid handler was used to dispense first strand synthesis reagents (0.4 µl) and second strand synthesis reagents mix (6 µl). cDNA from 96 different primers was pooled together, and 0.8-fold AMPure beads (0.8-fold ratio; Beckman Coulter) were used for cDNA clean up before the in vitro transcription. The RNA from the first round of IVT was purified by 0.8-fold RNAClean XP beads (0.8-fold ratio) and was used as a substrate for another round of first strand synthesis at 42°C for 2 hr using second-round primers (second Round Primer: NNNNNN), RNA:DNA hybrids were digested with RNase H (Thermal Fisher #18021014) at 37°C for 30 min. The second-round second-strand synthesis was carried out at 16°C with a T7-RA5 primer (GCCGGTAA TACGACTCACTATAGGGAGTTCTACAGTCCGACGATC). The resulting cDNA underwent another final second round IVT step at 37°C overnight and followed ExoSAP treatment (Affymetrix 78200) for 15 min at 37°C. Other steps were performed as described in the CEL-Seq2 protocol. Each single cell was sequenced on an Illumina Nextseq 550 sequencing system at a depth of ~0.5 million reads.

## Immunohistochemistry

Immunohistochemistry was performed on 3–5 days old flies. Flies were entrained in LD conditions for 3 days before being fixed with 4% (vol/vol) paraformaldehyde with 0.5% Triton X-100 for 2 hr and 40 min at room temperature. Brains were dissected with standard protocol in 0.5% PBST. The brains then washed twice (10 min) in 0.5% PBST buffer and blocked overnight in 10% Normal Goat Serum (NGS; Jackson ImmunoResearch Lab) at 4°C. The brains were then incubated in rabbit anti-PER at 1:1000 dilution, a rat anti-TIM at 1:200 dilution, a mouse or chicken anti-GFP antibody at a 1:1000, a rat anti-RFP antibody at 1:200, or a mouse anti-PDF antibody at 1:1000 for overnight, then the brains then washed twice (10 min) in 0.5% PBST buffer. The corresponding secondary antibodies were added and incubated for overnight. Brains were mounted in Vectashield (Thermal Fisher) and imaged on a Leica SP5 confocal microscope.

## Pre-processing of scRNA-seq data

scRNA-Seq data aligned to the *Drosophila* genome (dm6) using zUMIs (*Parekh et al., 2018*), together with STAR with default setting (*Dobin et al., 2013*). Only the alignments to annotated exons were used for UMI quantitation. Next, we filtered out low-quality cells using the following criteria: (1) fewer than 1000 or more than 6000 detected genes (where each gene had to have at least one UMI aligned); (2) fewer than 6000 or more than 75000 total UMI; (3) gene expression entropy smaller than 5.5, where entropy was defined as *-nUMI * ln(nUMI)* for genes with n*UMI >0*, where nUMI was a number of UMI in a cell.

We reported a relative, normalized number of UMIs in a cell as TP10K – transcripts per 10 thousand transcripts. Except where indicated specifically (*NPF* expression in LD), the reported gene expression on t-SNE plot is from LD and DD together.

## Dimensionality reduction and clustering

We integrated the data from six time points and LD and DD conditions using integration functions from Seurat 3 (version 3.0.2) package (*Butler et al., 2018*). First, we separately transformed data from each time point and condition using the normalization and variance stabilization of counts (SCTransform function in Seurat), regressing out numbers of genes, UMIs, detected genes per cell, sequencing batches, percentage of mitochondrial transcripts, and computing 3000 variable genes at each time point and condition. Next, we found a subset of variable genes that were common to six time points in LD and DD conditions. From this set of common variable genes, we removed the mitochondrial, ribosomal, and transfer RNA genes. The resulting, filtered set of genes was used for integrating data from all time points and conditions using Seurat *FindIntegrationAnchors* and *IntegrateData* functions using top 50 canonical correlation analysis (CCA) eigenvectors from canonical correlation analysis. Finally, we performed principal component analysis (PCA) on scaled gene expression vectors (z-scores) and reduced the data to the top 50 PCA components. We used graph-based clustering of the PCA-reduced integrated data with the Louvain method. We clustered the dimensionally reduced data using *FindNeighbors* and *FindClusters* functions in the Seurat package. We visualized the clusters on a two-dimensional map produced with t-distributed stochastic neighbor embedding (t-SNE).

## Differentially expressed genes in each cluster

We computed genes that were differentially expressed in each cluster using a negative binomial generalized linear model (Seurat *FindAllMarkers* function with a latent variable indicating time, sequencing batch, and condition). For each gene, the expression in a given cluster was compared with expression of cells in the remaining clusters. The p-values were adjusted for multiple hypothesis testing using Bonferroni method. We used an adjusted p-value significance of 0.05 and fold change cutoff of 1.25 as the threshold of significant differential expression.

## Identification of cycling transcripts

Cycling transcripts were identified using both the JTK cycle component of MetaCycle (*Wu et al., 2016*) and Fourier transformation (*Wijnen et al., 2006*). In MetaCycle, cycling transcripts in each cluster were identified by treating each individual cell as a replicate. Since Fourier transformation is designed for two replicates of circadian timepoints, the single cells at each timepoint were randomly split into two groups using a custom script and mean expression for each group was calculated resulting in two six time point datasets that were used for Fourier analyses. To be considered cycling, the following cutoffs were used: a JTK cycle Benjamini-Hochberg corrected q-value of less than 0.05, a F24 score of greater than 0.5, and a cycling amplitude (maximum expression divided by minimum expression) of at least 1.5-fold, and a maximal expression of at least 0.8 TP10K. If minimum expression was zero, transcripts passed amplitude cutoffs if they met expression requirements.

## Gene ontology analysis

Gene ontology analyses were performed using g:Profiler (*Raudvere et al., 2019*). All gene lists were compared to a custom background list that consisted of all genes expressed in the Clk856-GAL4 neuron clusters at a value of at least 0.8 TP10K. Separate background corrections were used for LD and DD conditions. A g:SCS corrected p-value of less than 0.05 was used as a cutoff for significance.

## Acknowledgements

We thank the Bloomington *Drosophila* Stock Center and Dr. Yi Rao for fly stocks in this study. We thank the members of the Rosbash lab for thoughtful discussion and comments. We greatly appreciate Drs. Orie Shafer, Sejal Davla, Dragana Rogulja, Bryan Song, Paul Garrity, Sebastian Kadener, and Fang Guo for insightful comments on the manuscript.

## Additional information

### Funding

| Funder | Author |
|---|---|
| Howard Hughes Medical Institute | Dingbang Ma<br>Dariusz Przybylski<br>Katharine C Abruzzi<br>Matthias Schlichting<br>Qunlong Li<br>Xi Long<br>Michael Rosbash |

The funders had no role in study design, data collection and interpretation, or the decision to submit the work for publication.

### Author contributions

Dingbang Ma, Investigation, Visualization, Methodology, Writing - original draft; Dariusz Przybylski, Katharine C Abruzzi, Formal analysis, Writing - review and editing; Matthias Schlichting, Methodology, Writing - review and editing; Qunlong Li, Xi Long, Methodology; Michael Rosbash, Conceptualization, Resources, Funding acquisition, Project administration, Writing - review and editing

### Author ORCIDs

Dingbang Ma (iD) https://orcid.org/0000-0002-5575-7604
Katharine C Abruzzi (iD) http://orcid.org/0000-0003-3949-3095
Matthias Schlichting (iD) http://orcid.org/0000-0002-0822-0265
Xi Long (iD) http://orcid.org/0000-0002-0268-8641
Michael Rosbash (iD) https://orcid.org/0000-0003-3366-1780

### Decision letter and Author response

Decision letter https://doi.org/10.7554/eLife.63056.sa1
Author response https://doi.org/10.7554/eLife.63056.sa2

## Additional files

### Supplementary files

• Source data 1. List of identified marker genes in each cluster. Table displaying the average log-fold change values, percentage of expression and p-value for the list of differentially expressed genes among clusters.

• Supplementary file 1. List of identified rhythmic genes in each cluster. Table displaying the minimum and maximum gene expression level, phase, and p-value for the list of rhythmically expressed genes among clusters in light:dark (LD) and constant darkness (DD) conditions.

• Supplementary file 2. Gene ontology (GO) term analysis of the cyclers in each cluster. Table displaying p-value for the GO term analysis of the rhythmically expressed genes among clusters in light:dark (LD) condition.

• Supplementary file 3. Gene ontology (GO) term analysis for all cyclers. Table displaying the GO terms and their p-values of the pooled rhythmically expressed genes.

• Transparent reporting form

### Data availability

The single-cell RNA sequencing data has been deposited in GEO under the accession code GSE157504. Code used in this analysis has been deposited in GitHub (https://github.com/rosbash-lab/scRNA_seq_clock_neurons), copy archived at swh:1:dir:19da0c557f8884812d196e912af760ba5f5ffe2b.

The following dataset was generated:

| Author(s) | Year | Dataset title | Dataset URL | Database and Identifier |
|-----------|------|---------------|-------------|-------------------------|
| Ma D, Przybylski D, Abruzzi KC, Schlichting M, Li Q, Long X, Rosbash M | 2021 | A transcriptomic taxonomy of *Drosophila* circadian neurons around the clock | https://www.ncbi.nlm.nih.gov/geo/query/acc.cgi?acc=GSE157504 | NCBI Gene Expression Omnibus, GSE157504 |

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
