## [Decision Letter]

**Acceptance summary:**

This manuscript reports single cell sequence analysis of the *Drosophila* clock neuron network at different times of day. The authors were able to cluster the neurons sequenced and line up the clusters with the known classes of clock neurons in the *Drosophila* brain. The data obtained will provide a very valuable resource for *Drosophila* circadian biologists and more generally for clock researchers seeking to dissect mechanisms by which clocks control outputs. Also, insights into the heterogeneity of neurons in the *Drosophila* brain are broadly relevant.

**Decision letter after peer review:**

Thank you for submitting your article "A transcriptomic taxonomy of *Drosophila* circadian neurons around the clock" for consideration by *eLife*. Your article has been reviewed by three peer reviewers, and the evaluation has been overseen by a Reviewing Editor and Utpal Banerjee as the Senior Editor. The reviewers have opted to remain anonymous.

The reviewers have discussed the reviews with one another and the Reviewing Editor has drafted this decision to help you prepare a revised submission.

The reviewers found your work of considerable interest, and did not recommend any additional experiments. However, they requested clarification and further analysis or discussion of some findings. We would be happy to publish a revised manuscript that addresses these issues.

This manuscript reports single cell sequence analysis at different times of day of the *Drosophila* clock neuron network. The authors were able to cluster the neurons sequenced and line up the clusters with the known classes of clock neurons in the *Drosophila* brain. The data obtained will provide a very valuable resource for *Drosophila* circadian biologists and more generally for clock researchers seeking to dissect mechanisms by which clocks control outputs.

Essential revisions:

1) The authors report that the number of cells/cluster correlates with higher clock gene expression and also with more cyclers. Could this reflect that the averaging in clusters with more cells is more efficient, which could therefore bias the analysis of rhythms? One way to check this would be to see if the numbers change when an equal number of cells is sampled from each cluster. This finding also raises the possibility that cycling genes could be missed in smaller clusters. This potential limitation should be discussed.

2) "Most striking is the cluster-specific transcription factor expression in the DNs and LPNs (labeled in yellow)."

Is there a way to show how this looks at the single cell level? In other words, how large are the fluctuations across individual cells (in terms of SD, not SEM)? What is the origin of this variability, is it technical? are there transcriptional fluctuations?

3) In the gene set analyses, what are the peak times for the functions listed? Are there some patterns? To link those findings with function, it might be insightful to stratify the GO analyses in time intervals.

4) Given that this is a resource article, important for the descriptive information it provides, reviewers recommend that the authors refrain from speculation in the Abstract ( "We suggest that these molecules orchestrate temporal regulation of synapse formation and/or strength").

5) Why is it that the number of sequenced cells does not roughly reflect the size of each specific cluster despite the use of a comprehensive clock driver (for instance, compare the number of cells labeled as small or large LNvs, two similarly sized clusters; or those labeled as DN1a or DN2)?.

6) There appears to be considerable heterogeneity within a cluster and maybe even within a cell. For instance, AstC-R2 is in ~50% of cells in clusters 9 and 12, even though cluster 12 is purported to represent a single LNd. There should be some discussion of the heterogeneity observed.

7) Given the complexity of the DN1p group, it would help to have a panel reporting the genes used to identify each DN1p group and the number of cells in each group. This might also indicate how many DN1ps are accounted for.

---

## [Author Response]

Essential revisions:1) The authors report that the number of cells/cluster correlates with higher clock gene expression and also with more cyclers. Could this reflect that the averaging in clusters with more cells is more efficient, which could therefore bias the analysis of rhythms? One way to check this would be to see if the numbers change when an equal number of cells is sampled from each cluster. This finding also raises the possibility that cycling genes could be missed in smaller clusters. This potential limitation should be discussed.

This is a good point, and the reviewer is correct that cycling genes could be missed – are almost certainly missed – in the smaller clusters. With only ~3-4 cells per timepoint, cell variation (including perhaps the inability to detect a particular transcript in a cell) can make much if not most cycling there hard to detect. We have added this caveat to our manuscript.

In addition, JTK cycle analysis is done using individual single cell transcript values rather than averages. This allows for the variation from cell-to-cell to be considered when evaluating cycling and for example prevents a single cell with a high value from dominating or distorting cycling.

To test this concept, we did multiple cycling analyses after random down sampling of one of the largest clusters into virtual clusters ranging from 25 to 200 cells. As expected perhaps, the number of identified cyclers increases with cell number, which supports this small cluster-large cluster distinction.

Nonetheless, the number of cyclers per cluster is not strictly dependent upon cell number, indicating that there are still some robust conclusions that can be drawn. First, differences between clusters remain even after accounting for the number of cells per cluster. There are 3 different clusters with approximately the same number of cells but with an order of magnitude difference in their number of cycling transcripts. Second, if we down sample the top clusters to only 90 cells each (as suggested by the reviewer), substantially different numbers of cyclers are still found in each cluster. We have added these points to the text.

2) "Most striking is the cluster-specific transcription factor expression in the DNs and LPNs (labeled in yellow)."Is there a way to show how this looks at the single cell level? In other words, how large are the fluctuations across individual cells (in terms of SD, not SEM)? What is the origin of this variability, is it technical? are there transcriptional fluctuations?

We thank the reviewer for this suggestion. To address this question, we performed an additional analysis and now include a violin plot of TF expression (please see Figure 5—figure supplement 1C).

For a subset of the transcription factor transcripts with robust time-of-day oscillations (e.g. *ham*, *gl* and *pdm3*), their variable expression is almost certainly biological. For other TFs, future studies will be necessary to address this issue.

3) In the gene set analyses, what are the peak times for the functions listed? Are there some patterns? To link those findings with function, it might be insightful to stratify the GO analyses in time intervals.

Thank you for this suggestion. We had this same thought and have explored the GO terms of cyclers that peak at specific times during the day. We tried both dividing the day into smaller 4-hour bins as well as simply looking at day (or putative day) and night. Looking in smaller time windows was less productive due to the smaller numbers of cyclers, however, simply separating day and night does give some insights. For example, in Figure 6, G-protein coupled receptor signaling is found as a significant GO term only during the night in both LD and DD datasets, while cell-cell adhesion and neuron projection morphogenesis are significant terms only in the day. We have added this information to the text as well as to Supplementary file 3.

4) Given that this is a resource article, important for the descriptive information it provides, reviewers recommend that the authors refrain from speculation in the Abstract ("We suggest that these molecules orchestrate temporal regulation of synapse formation and/or strength").

We have done as recommended.

5) Why is it that the number of sequenced cells does not roughly reflect the size of each specific cluster despite the use of a comprehensive clock driver (for instance, compare the number of cells labeled as small or large LNvs, two similarly sized clusters; or those labeled as DN1a or DN2)?.

As mentioned in our paper, single cell suspension preparation likely contributes to these differences. We used flamed rounded tips to triturate the papain treated brains, l-LNvs have dense arborizations to the optical lobe in the brain, which probably makes more of these neurons damaged during trituration with reduced recovery by FACS It is also possible that variation in cell size causes some neurons to be lost in FACS. Then of course there is the simple answer: “we don’t know.”

6) There appears to be considerable heterogeneity within a cluster and maybe even within a cell. For instance, AstC-R2 is in ~50% of cells in clusters 9 and 12, even though cluster 12 is purported to represent a single LNd. There should be some discussion of the heterogeneity observed.

Thank you for the comment. AstC-R2 expression is cycling within 24 hours in cluster 9 and 12 (please see Supplementary file 1). Since the clustering contains individual cells from all timepoints, it follows that some cells will have high levels and others low levels.

For other non-cycling transcripts, heterogeneity can also be technical. If a transcript is not very abundant, it can sit on the threshold of detectability; it can therefore be detected in one cell but not another. Indeed, the total number of transcripts identified per cell also varies presumably as a consequence of the efficiency of reverse transcription in that particular single-cell reaction.

7) Given the complexity of the DN1p group, it would help to have a panel reporting the genes used to identify each DN1p group and the number of cells in each group. This might also indicate how many DN1ps are accounted for.

We thank the reviewers for this suggestion and have incorporated a summary table of those DN1p neuron clusters with number of cells that were recovered in LD and DD conditions, the genes we used for DN1p identification is also listed (please see Figure 4—figure supplement 1D). We included all the identified marker genes in Source data 1.